# LayerDAG: A Layerwise Autoregressive Diffusion Model of Directed Acyclic Graphs for System

Mufei Li*, Viraj Shitole*, Eli Chien*, Changhai Man*
Zhaodong Wang†, Srinivas Sridharan‡ Ying Zhang†, Tushar Krishna*, Pan Li*
*Georgia Institute of Technology. *mufei.li@gatech.edu, viraj2@gatech.edu, ichien6@gatech.edu,*
*cman8@gatech.edu, tushar@ece.gatech.edu, panli@gatech.edu*
†Meta. *zhaodongwang@meta.com, zhangying@meta.com*
‡NVIDIA. *srisridharan@nvidia.com*

*Abstract*—Directed acyclic graphs (DAGs) are ubiquitous in the design and optimizations of systems. For example, neural networks have become a key computational workload for system design, and neural architectures are natively DAGs. Intermediate representations in compilers or hardware-synthesis tool to characterize execution dependencies and dataflows of computation also often take the form of DAGs. For sensitive scenarios, we believe that *learning* a conditional generative model of DAGs allows releasing synthetic data that preserves downstream utility while protecting intellectual property (obfuscation). In addition, such models can efficiently search the space of valid DAGs for desired properties, which is of great potential use to applications like compiler optimization. However, generating realistic DAGs is challenging due to their inherent directional and logical dependencies. This paper introduces LayerDAG, an autoregressive diffusion model designed to address these challenges in DAG generation. By iteratively removing the nodes without predecessors and their outgoing edges, we can obtain a unique tokenization that turns a DAG into a sequence of directed bipartite graphs and its nodes into a sequence of node layers. LayerDAG leverages autoregressive generation to model directional dependencies and employs diffusion models to capture logical dependencies within each bipartite graph. Empirical studies demonstrate that LayerDAG outperforms existing DAG generative models, particularly for generating large-scale DAGs with up to 400 nodes—a critical scenario for system. Our implementation will be available at https://github.com/Graph-COM/LayerDAG.

## I. Introduction

Directed acyclic graphs (DAGs) have broad applications in ML system design. With directed edges, they can naturally depict dataflows between operators in neural network model architectures, enabling neural architecture search [8]. Further, by treating subtasks as nodes and task dependencies as edges, DAGs provides a compact representation for optimizing the scheduling of task execution [31], compiler optimizations for hardware accelerators like TPUs [1], [27], and for optimizing circuit netlists [9].

There is a growing interest in leveraging AI models for AI/ML system design/optimization [12], [16], [29], and was in fact a key thrust of the Architecture2.0 Workshop. This necessitates learning the structure of DAG datasets corresponding to the underlying design-space (e.g., hardware design, compiler optimization, and so on) and generating representative DAGs

for the optimized solution. To this end, this work presents a generative model for synthesizing DAGs. We believe a generative model for DAGs serves several purposes. First, by learning the distribution of DAGs, a generative model allows sampling realistic synthetic DAGs that reflect the real data distribution. Second, in sensitive scenarios like system and hardware design, this enables preserving data utility while protecting intellectual property (**obfuscation**) [15], [32], [44]. Third, a conditional DAG generative model would be capable of efficiently searching the space of valid DAGs for property optimization [43]. For instance, consider distributed training of a foundation model with data parallelism [7]; replicas of a model can perform similar operations on a huge number of GPUs, and the model itself often consists of repetitive layers. This would make an execution trace of the program [21], [32] extremely large due to redundant and repetitive patterns. In such cases, a generative model may learn to identify common components and compress traces by generating smaller synthetic ones [5].

Serving as an abstraction for flows and node dependencies, DAGs pose significant challenges in developing powerful and efficient generative models due to their intrinsic strong directional and logical dependencies, such as control flows, logic gates, and dimension requirements of matrix operations. These complexities are further magnified in large-scale DAGs, presenting a unique combination of challenges regarding both scale and logical rules.

Motivated by the aforementioned challenges, we present LayerDAG, a layerwise autoregressive diffusion model of DAGs. To tackle the directional logical dependencies, inspired by the strong capability of language models in modeling sequential logic, we propose autoregressive generation with the following tokenization. By iteratively removing the nodes without predecessors and their outgoing edges, we observe that a DAG induces a unique ordered partition of the nodes, which we refer to as layers (Fig. 1), along with a partition of the edges into bipartite graph structures. LayerDAG performs autoregressive generation at the granularity of layers while following the direction of edges. To model the node and edge dependencies, we adopt diffusion models that have achieved remarkable generation quality in various domains [28], [36].

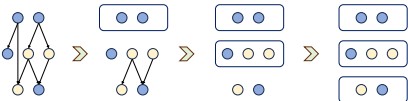

Fig. 1. Each DAG has a unique layerwise partition.

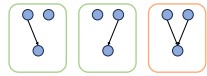

Fig. 2. The two edges are equally likely, but mutually exclusive. Naïvely predicting their existence by independently sampling from two Bernoulli distributions can violate the mutual exclusivity.

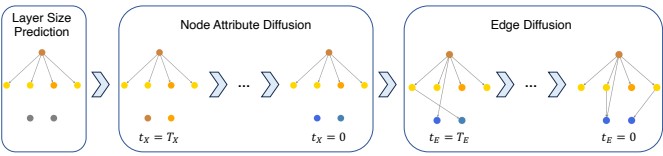

Fig. 3. Layerwise autoregressive diffusion.

Specifically, the diffusion model is conditioned on previously generated bipartite graphs to generate the next bipartite graph consisting of node attributes and edges. Essentially, our model is the first autoregressive diffusion model for DAG generation (Fig. 3).

Our model advances existing DAG generative models in multiple aspects. Methodologically, although autoregressive models have been adopted by D-VAE [43] and GraphP-NAS [18] for DAG generation, they treat either a single node or a node set of constant size as a token. This tokenization method imposes an order between nodes that should be incomparable in the partial order defined by the directed edges, violating the inductive bias inherent in the DAG structure. We argue that the violation may hurt the generalization capability of generative models. DiffusionNAG [2] employs diffusion models to generate only node attributes given a DAG structure. Diffusion models have been used to generate undirected graphs [14], [25], [36], but they ignore the directional information in DAGs, while our work demonstrates the necessity of the autoregressive component in modeling directional dependencies in DAGs. From the application perspective, all the existing works focus on generating small DAGs (with #nodes $\leq$ 24) for neural architecture search (NAS), while our model is capable of generating much larger flow graphs (up to $\sim$ 400 nodes) for system and hardware.

We conduct extensive experiments to verify the effectiveness of our model. To examine the model capability in learning strong directional logical rules, we construct a challenging synthetic dataset. To evaluate the conditional generation capability for the purpose of data sharing with obfuscation, we consider three real-world datasets – computational graphs on TPU, flow graphs on FPGA, and neural architectures deployed on edge devices. Each dataset contains thousands of DAGs, with individual DAGs comprising up to hundreds of nodes. On all four datasets, LayerDAG consistently achieves a better generation quality than the best baselines, by an average margin of 9.1%.

## II. BACKGROUND

Discrete Denoising Diffusion Probabilistic Model (D3PM) [3] is a diffusion model for discrete data. It has two phases, a forward diffusion process and a reverse process. Let $\mathbf{Z}^{(0)} \in \mathbb{R}^{M \times C}$ be the one-hot encodings of a categorical attribute with $C$ possible values for $M$ real samples. The forward process corrupts them into purely random samples by progressively changing the category of more samples over $T$ consecutive time steps. To corrupt $\mathbf{Z}^{(t)}$ into $\mathbf{Z}^{(t+1)}$, it computes and samples from a conditional distribution $q(\mathbf{Z}^{(t+1)}|\mathbf{Z}^{(t)}, t) = \mathbf{Z}^{(t)}\mathbf{Q}^{(t+1)}$, where $\mathbf{Q}^{(t+1)} \in \mathbb{R}^{C \times C}$ is a pre-determined transition matrix. A denoising network $\phi_\theta$ is trained to predict the uncorrupted data $\hat{\mathbf{Z}}^{(0)}$ from $(\mathbf{Z}^{(t)}, t)$. Composing the transition matrices across multiple time steps yields the closed-form expression $q(\mathbf{Z}^{(t+1)}|\mathbf{Z}^{(0)}, t) = \mathbf{Z}^{(0)}\overline{\mathbf{Q}}^{(t+1)}$, where $\overline{\mathbf{Q}}^{(t+1)} = \mathbf{Q}^{(1)}\mathbf{Q}^{(2)} \cdots \mathbf{Q}^{(t+1)}$. During the reverse process, the trained denoising network can be used to convert $\mathbf{Z}^{(T)}$ drawn from the prior distribution into realistic data.

DiGress [36] extends D3PM for non-autoregressive generation of undirected graphs with a single categorical node attribute. It treats the presence or absence of an edge between a node pair as a binary attribute. GraphMaker [19] extends DiGress for generating multiple categorical attributes.

## III. LAYERDAG

In this section, we present the LayerDAG framework. We first describe the unique layerwise DAG partition, which naturally leads to a layerwise tokenization for autoregressive DAG generation. To perform layerwise set-level predictions of node attributes and edges, we propose layerwise diffusion that is capable of modeling complex logical rules via refinement with multiple denoising steps. By adopting permutation equivariant and invariant model components, LayerDAG is also permutation invariant, which benefits model generalizability and robustness. As the cost of DAG generation is proportional to the constant number of denoising steps, we propose to alleviate the cost with a number of denoising steps proportional to the layer depth for a flexible quality-efficiency trade-off.

### A. Layerwise Partition and Tokenization

Consider an arbitrary DAG denoted by $(\mathcal{V}, \mathcal{E})$, where $\mathcal{V} = \{1, \cdots, N\}$ is the node set, and $\mathcal{E} = \{(u_e, v_e)\}_e$ is the set of directed edges. Assume the DAG has a set of nodes without predecessors, i.e., the nodes are not destinations of any edges, and we denote them by $\mathcal{V}^{(1)}$. Iteratively, we take $\mathcal{V}^{(l+1)} \subset \mathcal{V} \setminus \mathcal{V}^{(\leq l)}$ to be the set of nodes whose predecessors are in $\mathcal{V}^{(\leq l)}$, where $\mathcal{V}^{(\leq l)} = \bigcup_{i=1}^{l} \mathcal{V}^{(i)}$. It follows that $(\mathcal{V}^{(1)}, \mathcal{V}^{(2)}, \cdots, \mathcal{V}^{(L)})$ forms an ordered partition of $\mathcal{V}$ the moment $\mathcal{V}^{(L+1)} = \emptyset$ (Fig. 1), which we refer to as layers. For each layer depth $1 \leq l \leq L - 1$, we also take

$\mathcal{E}^{(l+1)} = \{(u_e, v_e) \in \mathcal{E} | u_e \in \mathcal{V}^{(\leq l)}, v_e \in \mathcal{V}^{(l+1)}\}$, and $(\mathcal{E}^{(2)}, \mathcal{E}^{(3)}, \cdots, \mathcal{E}^{(L)})$ forms an ordered partition of $\mathcal{E}$. Note that $(\mathcal{V}^{(\leq l)} \cup \mathcal{V}^{(l+1)}, \mathcal{E}^{(l+1)})$ is essentially a bipartite graph. This layerwise partition naturally extends to arbitrary node and edge attributes. The partition exists if $|\mathcal{V}^{(1)}| > 0$, which is guaranteed for a DAG. Furthermore, this way of construction is unique and allows reconstructing the original DAG from the sequence $(\mathcal{V}^{(1)}, \mathcal{V}^{(2)}, \mathcal{E}^{(2)}, \cdots, \mathcal{V}^{(L)}, \mathcal{E}^{(L)})$. The process of converting a raw data sample into a sequence is known as tokenization in the context of generative models, with notable examples being subwords for language models [30] and patches for image generation [26]. Essentially, layerwise partition leads to a layerwise tokenization method for DAGs.

### B. A Layerwise Autoregressive Generation Framework

Motivated by the layerwise tokenization, we propose a layerwise autoregressive framework that emulates the partition process for the purpose of DAG generation. Let $G^{(\leq l)}$ be a partially generated DAG with $l$ layers, and $G^{(\leq 0)} = G^{(0)}$ be the initial empty graph. To generate the $(l + 1)^{\text{th}}$ layer, it first predicts the number of new nodes with $p_\theta(|\mathcal{V}^{(l+1)}| \mid G^{(\leq l)})$. Then it generates the node attributes with $p_\theta(\mathbf{X}^{(l+1)} \mid G^{(\leq l)}, |\mathcal{V}^{(l+1)}|)$. Finally, it generates the edges with $p_\theta(\mathcal{E}^{(l+1)} \mid G^{(\leq l)}, \mathbf{X}^{(l+1)})$. The generation process terminates when a layer size of 0 is predicted.

### C. Layerwise Diffusion

The modeling of $p_\theta(\mathbf{X}^{(l+1)} \mid G^{(\leq l)}, |\mathcal{V}^{(l+1)}|)$ and $p_\theta(\mathcal{E}^{(l+1)} \mid G^{(\leq l)}, \mathbf{X}^{(l+1)})$ requires simultaneously sampling all set elements. Naïve sampling of set elements from multiple conditionally independent distributions cannot model the intra-set constraints like mutual exclusivity (Fig. 2). This is particularly an issue for generating real-world DAGs with heavy hard logical rules like systems and circuits, and the rule violations can accumulate over layers of generation.

To tackle this issue, we adopt discrete denoising diffusion for multiple rounds of set refinement. We use the empirical marginal distribution of a categorical attribute $\mathbf{m} \in \mathbb{R}^C$ as its corresponding prior distribution. The composed transition matrix is chosen to be $\overline{\mathbf{Q}}^{(t)} = \overline{\alpha}^{(t)} \mathbf{I} + (1 - \overline{\alpha}^{(t)}) \mathbf{1}\mathbf{m}^\top$, where $\overline{\alpha}^{(t)} = \cos^2\left(\frac{\pi}{2} \frac{t/T+s}{1+s}\right)$ is the cosine noise schedule [24], $\mathbf{I} \in \mathbb{R}^{C \times C}$ is the identity matrix, and $\mathbf{1} \in \mathbb{R}^C$ is the one-valued vector. We employ two separate diffusion processes for node attribute and edge generation (Fig. 3), as suggested by Li et al. [19]. For simplicity, we focus on categorical node attributes in this paper. Our approach can be extended to handle real-valued attributes [11], [37].

Autoregressive layerwise diffusion may be more efficient than non-autoregressive counterparts for DAG generation. Non-autoregressive diffusion models use a constant number of denoising steps in generation. A single denoising step involves refining the whole graph structure and node attributes, which is already very costly. In contrast, autoregressive layerwise diffusion only refines node attributes and edges for one layer in a single denoising step, and the total number of denoising steps naturally scales with respect to the number of DAG layers.

### D. Implementation

$p_\theta(|\mathcal{V}^{(l+1)}| \mid G^{(\leq l)})$, $p_\theta(\mathbf{X}^{(l+1)} \mid G^{(\leq l)}, |\mathcal{V}^{(l+1)}|)$, and $p_\theta(\mathcal{E}^{(l+1)} \mid G^{(\leq l)}, \mathbf{X}^{(l+1)})$ are shared across layers, and involve a DAG encoder. We use an off-the-shelf bidirectional message passing neural network (MPNN) [38], which allows efficient parallel training over DAGs and layer depths. A single bidirectional MPNN layer updates node representations with synchronous message passing over both the directed edges and their revered counterparts: $\sigma(\mathbf{A}\mathbf{H}\mathbf{W}_{\text{forward}} + \mathbf{A}^\top \mathbf{H}\mathbf{W}_{\text{reverse}} + \mathbf{H}\mathbf{W}_{\text{self}})$, where $\sigma$ is a non-linear layer, $\mathbf{A}$ is the adjacency matrix for a DAG, $\mathbf{H}$ is the node representation matrix, and $\mathbf{W}$'s are learnable weight matrices. Both layer size prediction $p_\theta(|\mathcal{V}^{(l+1)}| \mid G^{(\leq l)})$ and node attribute prediction $p_\theta(\mathbf{X}^{(l+1)} \mid G^{(\leq l)}, |\mathcal{V}^{(l+1)}|)$ involve computing graph representations with a set pooling operator like sum or mean over the updated node representations.

To generate $\mathbf{X}^{(l+1)}$, the node attribute prediction module first samples $\mathbf{X}^{(l+1, T_X)} \in \mathbb{R}^{|\mathcal{V}^{(l+1)}| \times C}$ from its prior distribution, where $T_X$ is the maximum number of denoising steps. Then iteratively, it samples $\mathbf{X}^{(l+1, t)}$ with a denoising network $\phi_{\theta_X}(G^{(\leq l)}, \mathbf{X}^{(l+1, t+1)}, t+1)$ for $t = T_X - 1, T_X - 2, \cdots, 0$. To predict the set of denoised node attributes, $\phi_{\theta_X}$ integrates the representations of $G^{(\leq l)}$ and $t + 1$ into the embeddings of $\mathbf{X}^{(l+1, t+1)}$, and then employs a set transformer, a transformer without positional encodings [35], over them for the final predictions. Similarly, the edge prediction module iteratively samples $\mathcal{E}^{(l+1, t)}$ with a denoising network $\phi_{\theta_E}(G^{(\leq l)}, \mathbf{X}^{(l+1)}, \mathcal{E}^{(l+1, t+1)}, t+1)$, which augments $G^{(\leq l)}$ by $(\mathbf{X}^{(l+1)}, \mathcal{E}^{(l+1, t+1)})$ for computing the node representations of $\mathcal{V}^{(\leq l+1)}$. To predict the probability of edge $(u, v) \in \mathcal{V}^{(\leq l)} \times \mathcal{V}^{(l+1)}$, it concatenates and transforms the representations of node $u$, node $v$, and $t + 1$ with an MLP.

Bidirectional MPNNs and transformers without positional encoding are permutation equivariant, while set poolings are permutation invariant. Consequently, all three modules are permutation invariant, and hence LayerDAG is also permutation invariant. In theory, sufficiently powerful permutation non-invariant models are capable of memorizing the training data without capturing the underlying directional and logical dependencies. Consequently, they may have much worse performance in generating unseen DAGs than LayerDAG (**generalizability**). This is particularly an issue for applications like obfuscation, where the goal is to release useful synthetic DAGs in replacement of the original DAGs.

### E. Layer-Index-Based Denoising Schedule

While refinement with multiple denoising steps improves the generation quality, the amount of time taken for generation also grows linearly in the number of denoising steps. Training multiple LayerDAG models with a variety of denoising steps does natively enable a trade-off between generation quality and efficiency, but is cumbersome and costly. A natural follow-up question is: Can we achieve flexible quality-efficiency trade-off with a single trained LayerDAG model?

As $l$ increases, both $|\mathcal{V}^{(\leq l)}|$ and $|\mathcal{E}^{(l+1)}|$ increase in general, resulting in more complex edge dependencies. To effectively

handle this pattern, we introduce a non-uniform denoising schedule for better allocation of the time budget. Specifically, we propose to set the total number of denoising steps to linearly increase in $l$.

$$T^{(l)} = T_{\min} + \lfloor (T_{\max} - T_{\min}) \cdot \min\{l/L_{\max}, 1\} \rfloor \quad (1)$$

where $T_{\min}$ and $T_{\max}$ are the minimum and maximum number of denoising steps to use during generation and allow users to make a flexible quality-efficiency trade-off. $l$ is the current layer index, $L_{\max}$ is the maximum number of layers in the training data, and $\lfloor \cdot \rfloor$ is the floor function.

## IV. RELATED WORK

GraphRNN [41] and Li et al. [20] propose autoregressive models to sequentially generate an undirected graph, one node/edge at a time. D-VAE [43] extends this approach to DAGs, and adopts topological orderings of nodes based on empirical studies. Although topological orderings respect the constraint of edge directions, they are not unique. Consequently, unlike LayerDAG, autoregressive models using topological orderings are not permutation invariant. For better efficiency, GRAN [22] proposes to sequentially generate node sets of a constant size and their incident edges, where the edges for the new node set are predicted simultaneously using a mixture of Bernoullis, potentially with intra-set connections.

## V. EXPERIMENTS

With empirical studies, we aim to answer the following questions. **Q1**) In terms of generation quality, how well does LayerDAG perform against the existing methods when applied to DAGs with hard logical rules and real-world applications in system and hardware design? **Q2**) Does layer-index-based denoising schedule enable an effective trade-off between quality and efficiency? If so, how well does LayerDAG perform against the existing methods with a comparable time budget?

### A. Datasets and baselines

**Datasets.** We construct a synthetic dataset of latent preferential DAGs (LP) with hard logical constraints including mutual exclusivity for layer size, node attributes, and edges, which allows directly evaluating the proportion of the generated DAGs that are valid (**validity**). Each node is associated with a binary label 0 or 1. The first layer of a DAG consists of a single node, whose label is drawn from a Bernoulli distribution with $p = 0.5$. Let $n_i^{(l)}$ be the number of nodes with attribute $i$ in layer $l$ for $i \in \{0, 1\}$. $n_i^{(l+1)}$ is sampled from $\mathcal{U}\left(\max\left\{0, n_i^{(l)} - (l+1)\right\}, n_i^{(l)} + (l+1)\right)$. Each new node can have one or two predecessors, and at least one of them belongs to the previous layer. In the case of two predecessors, they need to have distinct labels (**mutual exclusivity**).

In addition, we repurpose and adapt three representative real-world datasets, originally developed for DAG property prediction, to serve as testbeds for conditional DAG generation. The datasets are associated with computation workloads executed on diverse hardware platforms, and they well fit the end scenario of synthetic data sharing for system/hardware

TABLE I
DATASET STATISTICS. $|\mathcal{V}|$, $|\mathcal{E}|$, AND $L$ ARE AVERAGED OVER GRAPHS. $|\mathcal{V}^{(l)}|$ IS AVERAGED OVER LAYERS.

| Dataset | # graphs | $|\mathcal{V}|$ | max $|\mathcal{V}|$ | $|\mathcal{E}|$ | max $|\mathcal{E}|$ | $L$ | max $L$ | $|\mathcal{V}^{(l)}|$ | max $|\mathcal{V}^{(l)}|$ | # attributes |
|---|---|---|---|---|---|---|---|---|---|---|
| TPU Tile | 6, 301 | 40.8 | 394 | 42.9 | 711 | 11.2 | 72 | 3.6 | 21 | 1 |
| HLS | 2, 062 | 88.6 | 356 | 110.7 | 477 | 27.75 | 78 | 3.2 | 28 | 7 |
| NA-Edge | 2, 000 | 231.1 | 339 | 265.8 | 385 | 149.1 | 185 | 1.5 | 4 | 14 |

benchmarking. Originally released as part of the TpuGraphs dataset, **TPU Tile** is a collection of kernel graphs for machine learning workload on Tensor Processing Units (TPUs), with graph labels indicating the runtime averaged over a set of compilation configurations [27]. **High-level synthesis (HLS)** is a collection of data flow intermediate representation graphs for compiled C programs, with each DAG labeled according to the resource usage of look up table measured on Field Programmable Gate Arrays (FPGAs) [39]. **NA-Edge** is a collection of DAGs representing neural architectures, with labels indicating their inference latency on mobile CPU [8], [42]. Table I presents the dataset statistics.

Performing ground truth evaluations for conditional generation of DAGs in system and hardware design requires direct measurements on specific computational platforms. For example, the HLS dataset requires program implementation and measurement on FPGAs [39]. Such evaluations are computationally costly or infeasible due to limited access. Additionally, they demand specialized domain knowledge that often exceeds the expertise of general machine learning practitioners. Recently, employing ML-based surrogate cost models has emerged as a popular and effective alternative to direct measurement in various system and hardware optimizations [4], [6], [10], [13], [17], [23], [27], [33], [34], [45]. In light of these achievements, we propose to evaluate the quality of generated DAGs with ML-based surrogate models. Specifically, we partition the real labeled DAG datasets into training/validation/test subsets. After developing a DAG generative model using the training and validation subsets, we use the real training and validation labels as conditions for DAG generation. The generated labeled DAGs essentially form synthetic training and validation subsets. Inspired by previous practices [19], [40], we develop two DAG property prediction models based on bidirectional MPNN using the same automated pipeline respectively on the real and synthetic train/val subsets. We then compare the performance of the two models on the real test set using Pearson correlation coefficient. A generative model is considered better if its corresponding model achieves a performance closer to that of the model developed on the real subsets.

**Baselines.** GraphRNN, D-VAE, and GRAN were previously introduced in Section IV. Both GraphRNN and GRAN originally tackle the generation of non-attributed undirected graphs, and we extend them for attributed DAGs following the existing practices [18], [43], including the use of random topological orderings due to superior empirical performance. To compare the capability in set generation of LayerDAG against GRAN,

TABLE II
EVALUATION RESULTS. BEST RESULTS ARE IN **BOLD**. THE PEARSON
VALUE OBTAINED WITH THE REAL DATA IS ALSO LISTED FOR REFERENCE.

| Model | LP | TPU Tile | HLS | NA-Edge |
|---|---|---|---|---|
| | Validity ↑ | Pearson (Real: 0.761) | Pearson (Real: 0.976) | Pearson (Real: 0.996) |
| LayerDAG | **0.907** | **0.676** | **0.831** | **0.994** |
| D-VAE | 0.640 | 0.437 | 0.796 | 0.898 |
| GraphRNN | 0.367 | 0.619 | 0.751 | 0.989 |
| GRAN | 0.117 | 0.307 | 0.603 | 0.656 |

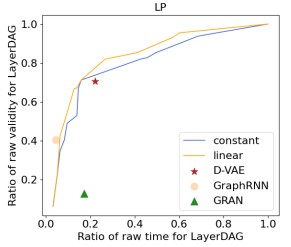 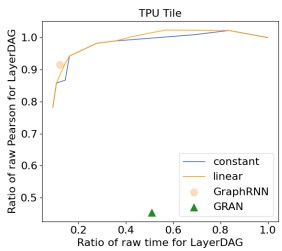

Fig. 4. Quality-efficiency trade-off. $x$-axis is the time budget divided by the raw generation time of LayerDAG. $y$-axis is the generation quality metric divided by the raw generation quality metric of LayerDAG.

we set the number of new nodes to generate in GRAN to be the averaged layer size, and extend mixture of Bernoullis into mixture of multinoullis for the set generation of node attributes. We extend all baselines and LayerDAG for conditional generation by augmenting the input data representations with sinusoidal embeddings [35] of the label.

### B. Evaluation for generation quality (Q1)

Table II presents the evaluation results. LayerDAG consistently outperforms all baselines across datasets, where the largest margin is for the LP dataset with hard logical rules. The experiment results demonstrate the aggregated benefit of permutation invariance and multiple rounds of refinement.

### C. Evaluation for layer-index-based denoising schedule (Q2)

Figure 4 presents the evaluation of layer-index-based denoising schedule in quality-efficiency trade-off on LP and TPU Tile. For an ablation study, we also experiment with a constant denoising schedule that simply utilizes a constant number of denoising steps smaller than the original number for all layers. Both schedules allow an effective quality-efficiency trade-off, while the generalized linear schedule often yield a better generation quality with the same time budget.

For comparison, we also evaluate the generation time of the baselines using the same batch size. When the GPU memory is insufficient, we use the largest feasible batch size instead. We plot all cases whose generation time is shorter than the original generation time of LayerDAG. Using layer-index-based denoising schedule, LayerDAG exhibits a better trade-off than most baselines except GraphRNN.

## VI. CONCLUSION

We propose LayerDAG, a layerwise diffusion model of DAGs. LayerDAG consistently outperforms the existing generative models of DAGs on four synthetic and real-world datasets. Utilizing a number of refinement rounds proportional to layer depth allows a better or comparable quality-efficiency trade-off compared with the baselines.

## ACKNOWLEDGEMENTS

We thank Haoyu Wang and Yinan Huang for their insights on dataset usage.

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
