# OpenReview forum: "LayerDAG: A Layerwise Autoregressive Diffusion Model of Directed Acyclic Graphs for System"
_iscaconf.org/ISCA/2024/Workshop/MLArchSys — MLArchSys 2024 OralPoster_

### Official Review · Reviewer_Qtsb · 2024-05-23
**Novel technique for generative models of DAGs, but hard to understand in some places**

**Confidence:** 3
**Rating:** 7

**Detailed Feedback And Questions For Authors:**

I really enjoy reading this paper. The paper explains most of the ideas and concepts clearly with illustrative figures and informative equations. As a non-expert in generative model or diffusion model, I find the paper not too hard to follow, and I learn more on how diffusion models work from reading the paper. As we have seen the success of a diffusion model in the image generation domain, applying this approach to DAGs is well motivated. However, it is non-trivial how to apply the diffusion model to a graph domain, in particular how to partition a graph into the units (tokens) that work well with the diffusion model. Thus, the proposed layer-wise tokenization idea is quite neat.

While most parts of the paper are easy to understand, some parts can be clarified. In particular, I don’t fully understand the experiment set up as a non-expert on diffusion models. First, the experiment said “we supply real labels to a model for generating synthetic training and validation subsets”. I assume that the labels refer to graph’s runtime/latency/resource usage. However, in order to train the diffusion model, we need the real graph data to train the generative model, not just the labels. Please clarify.

Next, “we develop two discriminative models”. What do you train these two models to do? And what does the test metric (pearson) measure? A correlation between the predicted values of the two discriminative models? What is the number for Pearson (real)? Please clarify.

In the technical section, the paper explains that the probability of edge (u, v) is predicted from the representation of node u, node v, t + 1. I assume that it would also depend on $\mathcal{E}$, but it is not there. Please clarify.

**Top Reasons To Accept The Paper:**

* Novel technique for generative models of DAGs
* The layer-wise “tokenization” idea is neat

**Top Reasons To Reject The Paper:**

* Missing some important details, making it hard to understand and evaluate

---

### Official Review · Reviewer_XntH · 2024-05-25
**Important problem, unclear evaluation metrics, missing related work**

**Confidence:** 4
**Rating:** 4

**Detailed Feedback And Questions For Authors:**

* How does LayerDAG compare with [Generative Flow Networks](https://arxiv.org/abs/2202.13903), [Graph Neural Networks](https://proceedings.mlr.press/v97/yu19a.html), and [Variational Autoencoders](https://arxiv.org/abs/1904.11088), and [graph diffusion models](https://arxiv.org/abs/2302.02591) for learning a distribution over DAGs?
* What does *validity* and *pearson* mean in Table 1?
* What is the intuition behind equation 1? Why do we need more denoising steps as we go deeper into the DAG? It seems like it's not making a significant difference looking at Figure 4.
* I didn't understand what you mean by *bipartite graph structure* of edges.
* Why *autoregressive layerwise diffusion may be more efficient than non-autoregressive counterparts for DAG generation*? I understand that it operates on smaller objects, but it has to do many sequential generation steps (one per layer).
* What is a *set transformer*? Is it a transformer without positional embeddings?
* For conditional generation, you augmented the baselines with sinusoidal embeddings of the label? Why did you use sinusoids? As far as I know, they're only used for positional embeddings. Why not give the label directly?

**Top Reasons To Accept The Paper:**

This paper proposes an algorithm for generative modeling of DAGs. DAGs are widely used in systems, e.g., for representing dataflows, jobs, circuits, etc.

**Top Reasons To Reject The Paper:**

1. It ignores a body of related work and baselines.
2. The evaluation metrics are not clearly defined.

---

### Official Review · Reviewer_t1vb · 2024-05-28
**LayerDAG: A Layerwise Autoregressive Diffusion Model of Directed Acyclic Graphs for System**

**Confidence:** 2
**Rating:** 6

**Detailed Feedback And Questions For Authors:**

Thank you for the submission to MLArchSys. The paper addresses a timely problem, offering a model that enables researchers to generate datasets for system optimizations. I have some suggestions/questions to may enhance the clarity of the paper:

1. Could the authors provide more insights into the specific advantages that each optimization in LayerDAG offers over other baselines across different datasets? Currently, the results are presented in aggregate.

2. Provide more details for the layer-based linear denoising schedule, explaining how it achieves superior generation quality within the same time constraints.

**Top Reasons To Accept The Paper:**

1. The paper presents an autoregressive diffusion model for generating Directed Acyclic Graphs (DAGs). The motivation for developing this model stems from the increasing use of DAGs in machine learning (ML) system design, where they can depict dataflows in neural networks, optimize task scheduling, and more. This is a timely prblem.

2. The flexible denoising schedule proportional to layer depth allows for a trade-off between generation quality and efficiency.

3. Comprehensive evaluation using different datasets.

**Top Reasons To Reject The Paper:**

1. The paper could provide more details on how the individual optimizations that are proposed for instance, the how the layer-based linear denoising schedule works for better generation quality with the same time budget.

---

### Decision · Program_Chairs · 2024-05-30

**Decision:**

Accept (Oral/Poster)

**Comment:**

Congratulations! We are pleased to inform you that your paper has been accepted for presentation at MLArchSys 2024. We look forward to your participation at the workshop. Further details regarding the schedule and format will be provided soon. See you at the workshop!